# Screening and Identification of ssDNA Aptamers for Low-Density Lipoprotein (LDL) Receptor-Related Protein 6

**DOI:** 10.3390/molecules28093838

**Published:** 2023-04-30

**Authors:** Xiaomin Zhang, Ge Yang, Wenjing Liu, Qing Liu, Zhuoran Wang, Kelong Fan, Feng Qu, Yuanyu Huang

**Affiliations:** 1School of Life Science, Advanced Research Institute of Multidisciplinary Science, Key Laboratory of Molecular Medicine and Biotherapy, Key Laboratory of Medical Molecule Science and Pharmaceutics Engineering, Beijing Institute of Technology, Beijing 100081, China; 2CAMS Key Laboratory of Antiviral Drug Research, Beijing Key Laboratory of Antimicrobial Agents, NHC Key Laboratory of Biotechnology of Antibiotics, Institute of Medicinal Biotechnology, Peking Union Medical College, Chinese Academy of Medical Sciences, Beijing 100050, China; 3Beijing Key Laboratory of Drug Resistance Tuberculosis Research, Beijing Tuberculosis and Thoracic Tumor Research Institute, Beijing Chest Hospital, Capital Medical University, Beijing 101125, China; 4CAS Engineering Laboratory for Nanozyme, Key Laboratory of Protein and Peptide Pharmaceutical, Institute of Biophysics, Chinese Academy of Sciences, Beijing 100101, China

**Keywords:** LRP6, aptamer, capillary electrophoresis, SELEX

## Abstract

Low-density lipoprotein receptor-related protein 6 (LRP6), a member of the low-density lipoprotein receptor (LDLR) family, displays a unique structure and ligand-binding function. As a co-receptor of the Wnt/β-catenin signaling pathway, LRP6 is a novel therapeutic target that plays an important role in the regulation of cardiovascular disease, lipid metabolism, tumorigenesis, and some classical signals. By using capillary electrophoresis–systematic evolution of ligands by exponential enrichment (CE-SELEX), with recombinant human LRP-6 as the target, four candidate aptamers with a stem-loop structure were selected from an ssDNA library—AptLRP6-A1, AptLRP6-A2, AptLRP6-A3, and AptLRP6-A4. The equilibrium dissociation constant *K_D_* values between these aptamers and the LRP6 protein were in the range of 0.105 to 1.279 μmol/L, as determined by CE-LIF analysis. Their affinities and specificities were further determined by the gold nanoparticle (AuNP) colorimetric method. Among them, AptLRP6-A3 showed the highest affinity with LRP6-overexpressed human breast cancer cells. Therefore, the LRP6 aptamer identified in this study constitutes a promising modality for the rapid diagnosis and treatment of LRP6-related diseases.

## 1. Introduction

The low-density lipoprotein receptor (LDLR) is a cell-surface glycoprotein, and it plays an important role in cholesterol metabolism by mediating the endocytosis process of low-density lipoproteins (LDLs). Cholesterol metabolism contributes to a great extent to cancer progression, including cell proliferation, migration, and invasion [1]. Low-density lipoprotein receptor-related protein 6 (LRP6) is a Type I transmembrane protein, and as a member of the LDLR family, it has a unique structure and a ligand-binding function. LRP6 is an essential Wnt co-receptor that can be involved in cell growth, proliferation, and differentiation processes by signaling through the plasma membrane and activating the Wnt/β-catenin signaling pathway [2,3,4,5,6]. The binding of cell-surface receptors LRP6 and Frizzled (FZD) to Wnt can initiate a signaling cascade effect that leads to the transcription of Wnt target genes [7,8]. The aberrant regulation of the Wnt/β-catenin signaling pathway is associated with the pathogenesis of a variety of diseases, including cancer, cardiovascular disease, and coronary artery disease [8,9]. With regard to cancer, LRP6 expression is upregulated in several cancers, including breast cancer [10,11,12], hepatocellular carcinoma [13], colorectal cancer [12,14], and prostate cancer [15,16]. By breast cancer tissue microarray assay, Zhang et al. [17] found that the percentage of patients with high LRP6 expression was 59.3% among 150 breast cancer patients. Research shows that LRP6 deficiency can inhibit the Wnt/β-catenin signaling pathway in triple-negative breast cancer (TNBC) cells, which subsequently inhibits the migration and invasion of TNBC cells [11], while the upregulation of LRP6 expression in metastatic prostate cancer will increase the risk of recurrence [18]. Furthermore, Tung et al. [13] showed that, compared to the corresponding non-tumorous livers and normal liver tissues, the transcript level of LRP6 was frequently up-regulated in human hepatocellular carcinomas (HCCs). In BEL-7402 HCC cells, the stable overexpression of constitutively active LRP6 can strengthen cell proliferation, cell migration and invasion in vitro, and tumorigenicity in nude mice. In addition, the results of Elham et al. [19] showed that, in 61% of sporadic colorectal cancer (CRC) patients, LRP6 was significantly upregulated in malignant tissues. The overexpression or activation of LRP6 could increase Wnt/β-catenin signaling in colorectal cancer cells, which promotes the metastasis of tumor cells in vitro [14]. In contrast, by inhibiting the expression of LRP6, miR-195 can downregulate the β-catenin signaling pathway and the transcription of target genes such as RUNX2 and VEGFa, which consequently reduces the metastasis of colorectal cancer [20]. In summary, as a potential target, the elevated expression level of LRP6 protein is strongly correlated with malignancy and the poor prognosis of tumors, and targeting LRP6 may provide a new approach for the treatment of complex diseases, such as cancer, atherosclerosis, and Alzheimer’s disease.

Based on SELEX technology, aptamers obtained from random oligonucleotide libraries are able to recognize targets and bind target molecules through intermolecular forces. Aptamers have high specificity and affinity comparable to that of antibodies, and their equilibrium dissociation constants can reach nanomolar or even picomolar levels. Moreover, compared to antibodies, aptamers also have numerous advantages [21,22], such as easy synthesis and modification, a wide range of target molecules, and low immunogenicity, which have shown promising applications in new drug development and disease diagnosis. The target molecule must be attached to a stationary support by a linker molecule in conventional SELEX technology [23,24,25]. However, the screening of aptamers with capillary electrophoresis–systematic evolution of ligands by exponential enrichment (CE-SELEX) occurs in a free solution, which allows the target to flow freely in the buffer, which increases the number of potential binding sites. Therefore, it is not necessary to prepare a stationary phase with the target attached. Typically, it takes only 1–4 rounds of screening in CE-SELEX to obtain aptamers with high specificity and high affinity from a random oligonucleotide library with a capacity of 10^13^–10^15^ [23,26,27,28]. There is currently no reported aptamer selection against LRP6.

Adopting the LRP6 protein as the target, four candidate aptamer sequences were obtained after four rounds of screening using the CE-based SELEX strategy. Afterwards, the affinities of the candidate aptamers were characterized, and multiple sequence comparisons and a conserved site analysis were performed. Moreover, candidate aptamers were molecularly docked with LRP6 protein, and then the combined model and interaction force between them were analyzed, which could be used to further validate their affinities. Finally, two low-*K_D_*-value candidate aptamers, AptLRP6-A1, and AptLRP6-A3, were selected, and their specific binding to target cells was verified at the cellular level. All results suggest that the LRP6-specific aptamers based on the CE-SELEX screening have the potential to be applied as a novel targeting tool for the diagnosis and treatment of diseases related to abnormal LRP6 protein expression.

## 2. Results and Discussion

### 2.1. Aptamers of LRP6 Screened through CE-SELEX

Since the human recombinant LRP6 used for aptamer screening is an Fc chimera, we introduced mouse IgG as a negative target in the first round of screening to eliminate sequences that bind to Fc fragments. First, 1 μmol/L oligonucleotides library (ssDNA_N40_) with the random sequence of 40 nt was interacted with 2 μmol/L mouse IgG. Unbound ssDNA_N40_ that did not bind to the mouse IgG was collected (the collection section was 5.9–6.5 min) as a secondary library for the next round of forward screening (Figure 1A). After template amplification, asymmetric PCR amplification, and ssDNA purification, sub-ssDNA1 was obtained for the second round of screening. In order to increase the screening pressure, the concentration of the target protein was reduced to 0.4 μmol/L LRP6 in the second round of screening. As shown in Figure 1B, the peak area of free sub-ssDNA1 decreased significantly after the addition of 0.4 μmol/L LRP6, and the complex peak appeared at 2.2 min, with a collection section of 2.0–2.8 min. The concentration of target protein LRP6 was further reduced to 0.2 μmol/L to repeat the screening procedure, and the third and fourth round secondary libraries (sub-ssDNA2 and sub-ssDNA3) were obtained (Figure 1C,D). After four rounds of screening, the final library was amplified and purified, and the recovered product was quantitatively detected to be 39.2 ng/μL, and then high-throughput sequencing was performed. Read block errors and primer polymers were eliminated from the sequencing results, and the four sequences from AptLRP6-A1 to AptLRP6-A4 with the highest frequency were selected (Table 1).

### 2.2. Predicting Structure and Motif of Candidate Aptamers

The secondary structures and thermodynamic parameters of the sequences of candidate aptamers AptLRP6-A1~AptLRP6-A4 were obtained using the NUPACK server, and M-fold server, respectively. Moreover, there are several stem-loop structures in the secondary structures of the four candidate aptamers (Figure 2), which may facilitate the binding to the target. Among the candidate aptamers, AptLRP6-A1 and AptLRP6-A3 had lower ∆G values (Table 1), indicating that they are more prone to spontaneous folding into secondary structures. More specifically, AptLRP6-A1, as the highest-frequency sequence in the high-throughput sequencing results, had the highest Tm value (62.5 °C) than the other three candidate aptamers, which also indicates that it forms a more stable secondary structure. Furthermore, the three-dimensional structures of the candidate aptamers were simulated using the 3dRNA/DNA tertiary structure prediction method, respectively.

The visualization of the sequence conservativeness based on multiple sequence comparisons of the candidate aptamers was obtained using the WebLogo 3 server [29,30]. More specifically, the total height at each sequence position in the generated sequence logos indicates the sequence conservation at the corresponding position, and the height of the nucleobase letters indicates their occurrence frequency. Among the candidate aptamer sequences (sites 1–40) without primers, C in columns 2 and 14, T in column 18, and G in columns 32 and 36 were the conserved nucleotides (Figure 3A). The MEME suite was used to analyze the types and numbers of motifs in the aptamer sequences. The maximum number of motifs was set to 3, and the minimum motif width and maximum motif width were set to 6 and 40, respectively. The motifs were obtained by the online server after 10 min of operation, and the results are represented by different colored squares (Figure 3B). Both AptLRP6-A1 and AptLRP6-A4 contained three motifs in their sequences, and the three motifs in AptLRP6-A1 were the “CAGGCAGCT” sequence starting at position +4, “CATTAGTCT” at position +15, and “CGGTATG” at position +34 (Figure 3C).

### 2.3. Molecular Docking Simulation

The binding of aptamers to LRP6 proteins is a process of molecular recognition involving intermolecular interaction forces such as hydrogen bonds, salt bridges, and hydrophobic interactions. Therefore, molecular docking, as one of the important components of computational biology, can be used to simulate the binding mode and affinity between ligands and receptors by computer. In the present work, the PLIP web server was applied to molecularly dock the LRP6 protein to the candidate aptamers AptLRP6-A1~AptLRP6-A4, and the optimal docking model was selected based on the binding free energy. Subsequently, Pymol software (https://pymolwiki.org/index.php/Windows_Install, accessed on 22 February 2023) was utilized to visualize the docking results. As a single transmembrane protein, the LRP6 receptor protein contains an extracellular domain, a transmembrane domain, and an intracellular domain. Among them, the extracellular domain consists of four YWTD type β-propellers, four EGF-like domains, and one LDLR type A domain [2], in which the YWTD type β-propellers are preferentially bound by different Wnt proteins and inhibitors [3,31,32,33]. Furthermore, the pair of the first β-propeller and EGF-like domain is termed E1, and the second to fourth domain pairs are termed E2-E4 sequentially in the nomenclature adopted by Cheng et al. [31,34].

Three types of interactions were observed in the complex composed of aptamer AptLRP6-A3 with LRP6E1E2, including seven hydrogen bonds, five salt bridges, and five hydrophobic interactions (Figure 4). Meanwhile, only two types of interactions, hydrogen bonds and salt bridges, were observed in the complexes formed by the candidate aptamers AptLRP6-A1 and AptLRP6-A2 with LRP6E1E2, respectively. Three types of interactions were present in the complex composed of the candidate aptamer AptLRP6-A4 with LRP6E1E2, which were 18 hydrogen bonds, 3 salt bridges, and 1 hydrophobic interaction, but the binding site of the EGF-like domain in the complex structure was involved in all three types of forces. Similarly, among the docking results of the candidate aptamer AptLRP6-A2 with LRP6E3E4 (Figure 5), there were 11 hydrogen bonds, 1 salt bridge, and 1 hydrophobic interaction of the aptamer with the EGF-like domain in the structure of LRP6E3E4. In contrast, there were only two hydrogen bonds and one salt bridge between the aptamers and the EGF-like domain in the docking results of the candidate aptamer AptLRP6-A3 with LRP6E3E4. Compared to the other two candidate aptamers, the relatively higher affinity candidate aptamers, AptLRP6-A1 and AptLRP6-A3, almost did not bind to the EGF-like domain, but they participated in the binding on the upper surface of the YWTD β-propeller structural domain in the LRP6 structure, which is consistent with the binding sites of Wnt proteins, inhibitors, etc., to the LRP6 protein.

Furthermore, the analysis results of the motif show that there were 13 nucleotides in the interaction of the candidate aptamer AptLRP6-A2 with the LRP6 protein, which were G2, A9, G10, A35, C42, C43, T44, G45, C46, G47, T48, A49, and A73, respectively, of which 8 belonged to the conserved nucleotides in the motif. Similar to the analysis results of the candidate aptamer AptLRP6-A4, 21 nucleotides interacted with the LRP6 protein, 14 of which belonged to nucleotides in the motif. In contrast, in the interaction of the candidate aptamer AptLRP6-A1 with LRP6 protein, only 3 nucleotides (C31, 59T, and 60G) out of 23 were located in the motif of the aptamer AptLRP6-A1. In the analysis results of the candidate aptamer AptLRP6-A3, only nucleotide G52 was a conserved nucleotide out of the 13 nucleotides in the interaction with the LRP6 protein. The above analysis results show that the nucleotides involved in binding the LRP6 protein in the two high-affinity candidate aptamers, AptLRP6-A1 and AptLRP6-A3, were mainly non-conserved nucleotides. Meanwhile, the nucleotides in the two low-affinity candidate aptamers, AptLRP6-A2 and AptLRP6-A4, were mainly conserved types of nucleotides in the sequence motif. Accordingly, it can be inferred that the more conserved nucleotides involved in binding target proteins in candidate aptamers, the lower their affinity may be. This also indicates the importance of motif analysis and molecular docking simulations in the selection of the best candidate aptamer.

Based on the molecular docking results, the sites in the four candidate aptamer sequences that bind to the LRP6 protein were selected for the mutation (Appendix A), and the affinity magnitude of the mutated candidate aptamers was determined by the SPR method. The results show that the stem-loop structure of the candidate aptamers was significantly altered after site-directed mutation (Appendix A). Furthermore, the *K_D_* values of the mutation candidate aptamers MutLRP6-A1, MutLRP6-A3, and MutLRP6-A4 were 11.3 μmol/L, 2.68 μmol/L, and 8.47 μmol/L, respectively (Appendix A), and they were much greater than the initial *K_D_* values of the candidate aptamers. In addition, AptLRP6-A2, which had the lowest affinity among the candidate aptamers, was no longer bound to the LRP6 protein after the site-specific mutation. The above results verify that the sites analyzed by molecular docking were the key binding sites.

### 2.4. Affinity Characterization of Candidate Aptamers

The affinities of four candidate sequences AptLRP6-A1~AptLRP6-A4 were characterized by capillary zone electrophoresis (CZE), and their equilibrium dissociation constants (*K_D_*) were calculated. In Figure 6A–D, the 0.2 μmol/L free candidate aptamers (AptLRP6-A1~AptLRP6-A4) peaked at about 7.6 min. After the introduction of 1 μmol/L LRP6, stable complex peaks of the candidate sequences with LRP6 appeared at 3.8 min, which demonstrates the good affinity of the candidate aptamers. Relatively few unbound candidate aptamers were still observed between 6.0 and 8.0 min of the CE chromatogram, which were sequences in dissociation states. The *K_D_* values of AptLRP6-A1~AptLRP6-A4 were 0.118 ± 0.019 μmol/L, 1.279 ± 0.279 μmol/L, 0.105 ± 0.018 μmol/L, and 0.661 ± 0.092 μmol/L, respectively. The results show that among the four candidate aptamers, AptLRP6-A1 and AptLRP6-A3 had the highest affinities, which were about 10 times higher than that of AptLRP6-A2 (Figure 6E). To further investigate the formation process of the aptamer–target protein complex, the candidate aptamer AptLPR6-A1, at a concentration of 0.2 μmol/L, was taken as an example, and then different concentrations of the LRP6 target protein were added to it. As shown in Figure 6F, the peak of AptLRP6-A1 appeared at 6.8 min, and the peak area of free AptLPR6-A1 gradually decreased after 0.01–0.5 μmol/L LRP6 was added. When the concentration of LRP6 increased to 0.05 μmol/L, the first detectable complex peak appeared at 3.1 min. Between the peaks of the complex of the free AptLPR6-A1, there was a considerable amount of AptLPR6-A1 in the dissociation state. It is noteworthy that the peak area of the complex increased with the increase in the protein concentration, indicating that the AptLPR6-A1 bound to the LRP6 in a concentration-dependent way and is expected to be developed as a potential target protein detection probe. CE chromatography accurately showed the dynamic process of the aptamer–target complex formation, which provides sufficient evidence for complex identification.

The surface plasmon resonance technique was used to verify the affinity between the LRP6 protein and the candidate aptamers. Since plasma waves propagate on metal surfaces, the metal surface-coupled ligand protein LRP6 caused a change in the resonance angle when interacting with the candidate aptamers. The *K_D_* value could be eventually calculated by monitoring the change in the resonance angle and recording the SPR signal at different candidate aptamer concentrations. The SPR signals were enhanced gradually with increasing concentrations of the candidate aptamers AptLRP6-A1~AptLRP6-A4 (Figure 7A–D). The *K_D_* value trends of the candidate aptamers determined by SPR were consistent with those determined by CE-LIF, and thus, validate the affinity of the candidate aptamers for LRP6 (Table 2). Additionally, four proteins, HSA, LDL, FZD8, and S100B, as the controls were reacted with different concentrations of AptLRP6-A3, and the results show no significant change in the SPR signal (Figure 7E,F), which verifies the specific binding of the candidate aptamers to the LRP6 protein.

Finally, the gold nanoparticle (AuNP) colorimetric method was also applied to further verify the specific binding of the candidate aptamers to the target proteins. With the increase in the candidate aptamer concentration, the number of candidate aptamers adsorbed on the AuNPs decreased, and the AuNPs’ state changed from dispersed to aggregated, accompanied by the solution changing from red to blue (Appendix A). In contrast, the color of the AuNPs remained unchanged after the co-incubation of several control proteins (HSA, LDL, FZD8, and S100B) with the candidate aptamer AptLRP6-A3 (Appendix A). This further verifies that the candidate aptamers were bound specifically to the LRP6 protein.

### 2.5. Binding Specificity of Candidate Aptamers to Target Cells

Based on the affinity assay results of the candidate aptamers, two candidate aptamers with the lowest *K_D_* values, AptLRP6-A1 and AptLRP6-A3, were selected to verify their binding specificity at the cellular level. Firstly, the Cy5-labeled candidate aptamers were co-incubated with LRP6-overexpressed MDA-MB-231 cells and LRP6-underexpressed U937 cells, respectively. Subsequently, imaging of the treated cells was performed using a confocal laser scanning microscope. The experimental results show that a distinct Cy5 red fluorescent signal could be observed around the MDA-MB-231 cell membrane when the target cells were co-incubated with the candidate aptamer AptLRP6-A3 (Figure 8A), while the fluorescent signal was diminished when the target cells were co-incubated with the candidate aptamer AptLRP6-A1. In addition, when the candidate aptamers AptLRP6-A1 and AptLRP6-A3 were co-incubated with U937 cells, there was almost no fluorescence signal observed (Figure 8B). This indicates that the candidate aptamer AptLRP6-A3 was able to bind to the target cells specifically at the cellular level, and the binding level was higher than that of aptamer AptLRP6-A1, which is consistent with the results of the candidate aptamer affinity determined by CE-LIF. To further investigate whether the candidate aptamers had potential toxic effects, the candidate aptamers AptLRP6-A1~AptLRP6-A4 were co-incubated with MDA-MB-231 cells and U937 cells for 24 h, respectively, and then the cell viability was calculated. The results show that the candidate aptamers did not affect cell proliferation significantly (Appendix A), and they are safe potential target molecules.

## 3. Materials and Methods

### 3.1. Reagents and Cell Lines

The ssDNA library (5′-AGCAGCACAGAGGTCAGATG-N40-CCTATGCGTGCTACCGTGAA-3′), forward primer (5′-AGCAGCACAGAGGTCAGATG-3′), reverse primer (5′-TTCACGGTAGCACGCATAGG-3′), and candidate aptamers AptLRP6A1~AptLRP6A4 were synthesized and purified by Sangon Biotech Co., Ltd. (Shanghai, China). A capillary electrophoresis instrument with a fluorescence detector (Beckman P/ACE MDQ) was purchased from Beckman-Coulter Co., Ltd. (Brea, CA, USA). Neutral-coated capillaries (75 μm inner diameter) were purchased from Handan Xinnuo Fiber Optic Chromatography Co., Ltd. (Handan, China). Recombinant Human LRP6 was purchased from Bio-Techne (R&D Systems, Minneapolis, MN, USA). Dulbecco’s Modified Eagle Medium (DMEM), RPMI medium 1640, and fetal bovine serum were purchased from Beijing BioDee Biotechnology Co., Ltd. (Gibco, Beijing, China). The human breast cancer cell line MDA-MB-231 and human histiocytic lymphoma cell line U937 were obtained from the American Type Culture Collection (ATCC, Manassas, VA, USA). The Cell Counting Kit-8 (CCK-8) was obtained from YEASEN Biotech Co., Ltd. (Shanghai, China). The purity specifications of all chemical reagents were analytical reagents.

### 3.2. CE-SELEX

A mixture was obtained by incubating the treated ssDNA nucleic acid library with the LRP6 protein. Subsequently, the mixture was isolated, analyzed, and collected using CE at a voltage of 20 kV and a temperature of 25 °C. In order to prepare the secondary nucleic acid library for the next round of screening, the collected complexes needed to be amplified by PCR. Finally, the PCR products obtained from the last round of amplification were entrusted to Sangon Biotech Co., Ltd. to perform high-throughput sequencing with the Illumina MiSeq sequencing platform. After removing the primer dimers, the sequence data from the sequencing results were sorted by frequency, and the candidate aptamer sequences were selected by frequency analysis.

For the CE-SELEX screening condition of the aptamers, the screening condition was divided into two parts due to the complex separation in a free solution. (a) An incubation buffer was used for the target protein interaction with the ssDNA library. An incubation buffer was the storage condition (pH 8.0, 0.155% glutathione, 0.395% Tris-HCl) of the target protein, under which the activity of the protein could be well maintained. In this state, the nucleic acid library interacted with the target protein to form a dynamic “binding–dissociation” equilibrium, and then the stable complex was separated. (b) Running buffer was used for the separation of the protein–ssDNA complex by capillary electrophoresis. In the CE-SELEX process, the target protein did not need to be immobilized. The equilibrium mixture of the ssDNA library and the target protein remained in a dynamic “binding–dissociation” equilibrium in the capillary-free solution after injection. The stable complex could be separated and collected due to the difference in the charge/mass ratio under the drive of the electric field. Therefore, the optimized capillary electrophoresis running buffers were required to achieve complex separation. The separation conditions we determined were 50 mmol/L borate buffer solution at pH 7.6, and the complex could be separated in about 3 min.

Protein–ssDNA complexes are formed under the conditions of optimal protein activity, and subsequent separation conditions may cause certain screening pressure on the stability of the complexes, which allows the aptamers with excellent affinity to remain. This is beneficial for aptamers to maintain good performance in different environments.

### 3.3. Structure and Motif Predictions

The secondary structures of the candidate aptamers were predicted and simulated using the NUPACK server (https://nupack.org/ (accessed on 22 February 2023)), and then the minimum free energy (MFE) proxy structure and equilibrium base-pairing probabilities were obtained at 37 °C. Moreover, the thermodynamic parameters of the candidate aptamers were predicted and analyzed using an M-fold server (http://unafold.rna.albany.edu/?q=mfold (accessed on 22 February 2023)). In addition, the 3D structures of the candidate aptamers were predicted using the 3dRNA/DNA tertiary structure prediction method (http://biophy.hust.edu.cn/new/3dRNA (accessed on 23 February 2023)). Finally, the lowest-energy 3D structure model 1 was selected from the predicted multiple simulations.

There are specific sequence fragments called motifs in nucleic acid sequences, which are closely related to the function of the sequence. Graphical representations of multiple sequence comparisons for the candidate aptamers without primers were obtained using WebLogo 3 (https://weblogo.threeplusone.com/ (accessed on 22 February 2023)), and the conserved sequence regions of the candidate aptamers without primers were predicted by the online tool MEME (http://meme-suite.org/ (accessed on 22 February 2023)) to further elucidate the role of motifs in binding target proteins, and thus, assist in the selection of the best aptamer sequence.

### 3.4. Binding Mode Predictions

The PDB files of the candidate aptamers AptLRP6-A1~AptLRP6-A4 were acquired using the 3dRNA/DNA tertiary structure prediction method (http://biophy.hust.edu.cn/new/3dRNA (accessed on 23 February 2023)), and the LRP6E1E2 (PDB ID: 3s94) [34] and LRP6E3E4 (PDB ID: 4a0p) [33] structures were obtained from the RCSB PDB database (https://www.pdbus.org/ (accessed on 22 February 2023)). In addition, the HNADOCK server (http://hdock.phys.hust.edu.cn/ (accessed on 22 February 2023)) was applied to predict the binding complex structure between the candidate aptamers and the LRP6 proteins based on a hierarchical docking algorithm of an FFT-based global search strategy and an intrinsic scoring function for nucleic acid interactions. The PDB files of the candidate aptamers and LRP6 proteins were submitted to the HNADOCK server (accessed on 23 February 2023), which took about 1 h to complete the docking process and generate the top 100 binding models (model 1~model 100). The resulting interface of this server also provided the ranks and docking scores of the top 10 complex models. Subsequently, the Protein-Ligand Interaction Profiler (PLIP) web tool [35] (https://plip-tool.biotec.tu-dresden.de/plip-web/plip/index (accessed on 23 February 2023)) was used to analyze the non-covalent interactions between the LRP6 proteins and the candidate aptamers. Finally, Pymol molecular visualization software (https://pymolwiki.org/index.php/Windows_Install (accessed on 22 February 2023)) was used to view and analyze the docking results.

### 3.5. Affinity Characterization

The affinity determination between the candidate aptamers and the LRP6 proteins was based on the nonequilibrium capillary electrophoresis of equilibrium mixtures (NECEEM) established by Krylov’s group [36,37], which was performed using the CE-laser-induced fluorescence (LIF) assay. The magnitude of the *K_D_* value was calculated according to the following equation [38], where [P0] and [DNA] denote the concentrations of LRP6 protein and ssDNA, respectively, and A1, A2, and A3 denote the peak areas of free ssDNA, dissociated regions, and complexes, respectively.
(1)$$K_{D}=\frac{\left[P0\right]\left(1+\frac{A1}{A2+A3}\right)-[DNA]}{1+\left(A2+A3\right)/A1}$$

Surface plasmon resonance (SPR) analysis was performed on a CM5 SPR chip integrated into the Reichert4 SPR system (Reichert^®^, Buffalo, NY, USA). All solutions were prepared using ultrapure water obtained from the Master Touch-S15UVF Pure Water Purification system. The running buffer used throughout the analysis was PBST buffer (pH 7.4, with 0.05% Tween-20), which was filtered through a 0.22 µm membrane filter before use. The target/control proteins were prepared with pH 4.5 sodium acetate solution at 200 μL and 0.25 μg/μL. The candidate aptamers were configured with 1% PBST to the desired gradient concentration. The analysis temperature was controlled at 25 ± 1 °C. The target/control proteins were immobilized on the surface of the gold film of the chip containing carboxymethyl glucan by covalent amine coupling. The CM5 SPR chip was composed of four channels; the fourth channel was set as the reference channel, and the other three channels were the sensing channels. The target/control proteins were immobilized on the sensing channel, and the reference channel was activated but not immobilized. Furthermore, ethanolamine was used as a blocker to fill in the blanks. The final signal from the sensing channel was normalized by subtracting the signal from the reference channel. The target/control proteins were injected into the respective sensing channels at a rate of 10 µL/min for 300 s, and the CM5 chip was ready with the immobilization of protein. Then, the candidate aptamers of the concentration gradient were successively injected into the SPR channels at a speed of 25 µL/min to collect the real-time SPR signal.

In addition, the affinities and specificities of the candidate aptamers were verified using the AuNPs colorimetric method. First, equal volumes of the AuNP solution and the candidate aptamer solution were added to the 96-well plate and incubated for 15 min at room temperature. Secondly, LRP6 protein, with different concentration gradients, was added and incubated for 20 min. Finally, a change in the AuNP color was observed after adding 0.9 mol/L NaCl solution.

### 3.6. Targeting of Aptamers to Cells

MDA-MB-231 cells and U937 cells (2 × 10^4^ cells/mL) were transferred to confocal dishes and incubated in an incubator at 37 °C and 5% CO_2_ for 24 h. Subsequently, Cy5-labeled AptLRP6-A1 and AptLRP6-A3 (200 nmol/L) were co-incubated with the cells for 30 min, respectively. An incubation temperature of 4 °C was used to avoid the non-specific adsorption of the candidate aptamers to the cells. After incubation, the cells were washed 3 times using PBS buffer and imaged by laser confocal microscopy to observe the binding and specificity of the candidate aptamers to the cells.

Furthermore, to explore whether the candidate aptamers affected cell growth, the CCK8 kit was used to detect the effect of the candidate aptamers on cell proliferation. MDA-MB-231 and U937 cells were inoculated into 96-well plates (2 × 10^4^ cells/mL, 100 μL), which were then placed into the incubator for a period of pre-culture. The candidate aptamers were added to each well separately at 1 μmol/L and incubated for 24 h. Finally, the CCK8 solution was added to each well at 10 μL for the reaction, and the absorbance at 450 nm was measured by a multimode microplate reader.

## 4. Conclusions

In the current study, four rounds of screening were performed in an efficient and rapid manner using the CE-SELEX screening method, and as a result, four candidate aptamers, AptLRP6-A1~AptLRP6-A4, were obtained successfully. Among them, AptLRP6-A1 had the minimum Gibbs free energy (−7.29 kcal/mol), the maximum Tm value (62.5 °C), and the most stable secondary structure. Meanwhile, the candidate aptamers AptLRP6-A1 and AptLRP6-A3 had relatively low *K_D_* values of 0.118 ± 0.019 μmol/L and 0.105 ± 0.018 μmol/L, respectively. The affinity and specificity of the candidate aptamers were further verified by SPR analysis and the AuNP colorimetric method. There were three non-covalent interaction forces between the nucleotides in the candidate aptamers and the amino acids in the LRP6 protein, which were hydrogen bonds, salt bridges, and hydrophobic interactions. Furthermore, most of the nucleotides bound to the LRP6 protein in the candidate aptamers AptLRP6-A1 and AptLRP6-A3 did not belong to nucleotides in the motif, while the opposite result was observed for the candidate aptamers AptLRP6-A2 and AptLRP6-A4. In addition, AptLRP6-A3 was able to specifically bind to the surface of the LRP6-overexpressing MDA-MB-231 cells, and thus, could be employed as a potential target molecule. In summary, the screened LRP6 aptamers are promising for further application in the diagnosis and treatment of diseases associated with aberrant LRP6 protein expression.

## Figures and Tables

**Figure 1 molecules-28-03838-f001:**
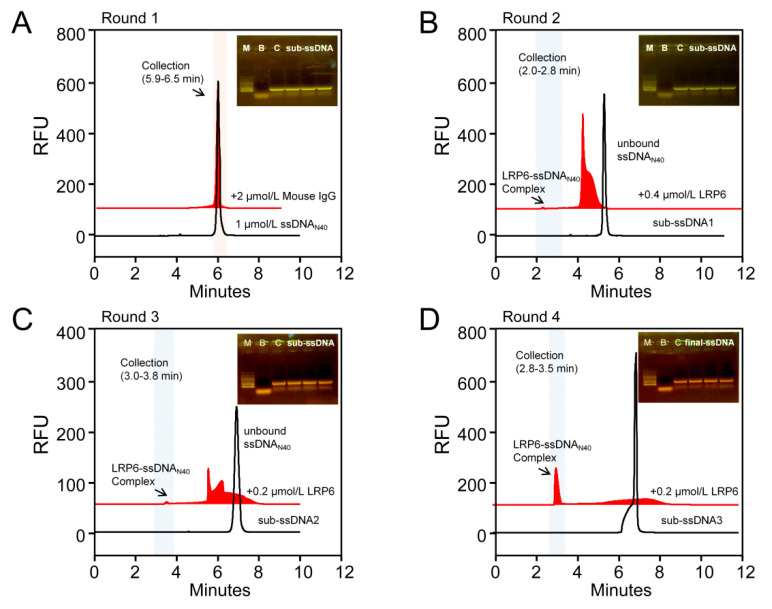
LRP6 aptamers were screened through CE-SELEX. (**A**–**D**) Rounds 1–4 of SELEX. The arrow points to the complex of LRP6 with the oligonucleotide library. The upper right corner of each diagram is a gel electrophoresis diagram of the PCR amplification of the collected components, where M represents the marker, B represents the blank, and C represents the ssDNA_N40_ control sample. Finally, there are three parallel samples of the PCR products of the collected complex components.

**Figure 2 molecules-28-03838-f002:**
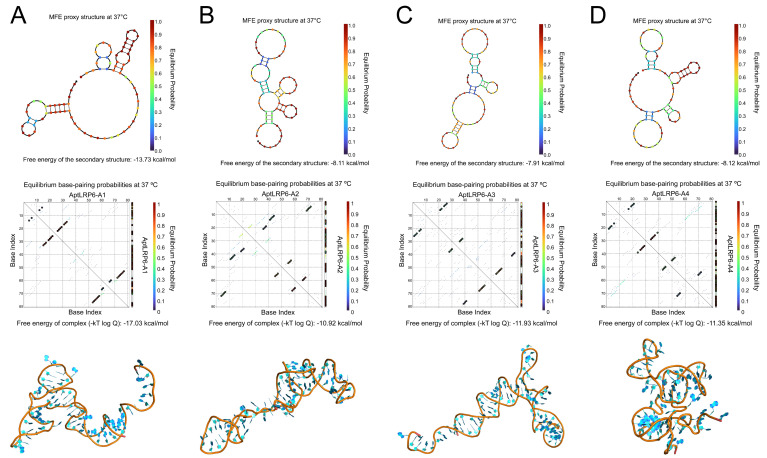
Secondary and tertiary structure prediction of candidate aptamers. (**A**–**D**) The secondary and tertiary structures of AptLRP6-A1~AptLRP6-A4 were predicted by NUPACK software (https://nupack.org/, accessed on 22 February 2023) and the 3dRNA/DNA tertiary structure prediction method, respectively. The second row shows the pair probabilities heat map, where the higher the hybridization probability (from blue to red), the more stable binding between bases.

**Figure 3 molecules-28-03838-f003:**
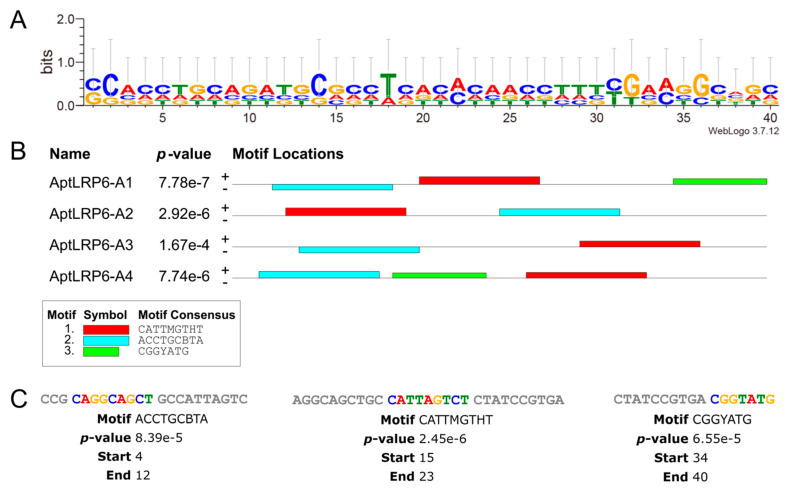
Motif prediction and analysis of candidate aptamers. (**A**) Sequence logo of candidate aptamers generated by the WeLogo 3 server. (**B**) Three different motifs of candidate aptamers were generated by the MEME Suite program. Each color square represents a motif. In this manuscript, the calculation of the number of motifs included the complementary strand of the sequence. (**C**) Motif sequence and specific location in AptLRP6-A1. The candidate aptamer sequences used for motif prediction and analysis did not contain forward primers or reverse primers.

**Figure 4 molecules-28-03838-f004:**
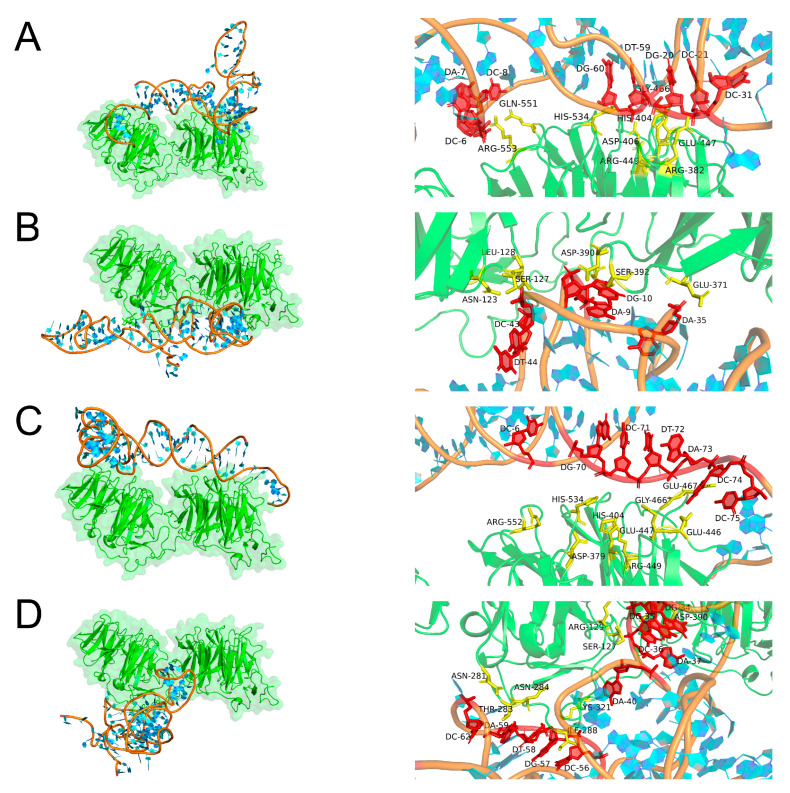
Docking prediction of candidate aptamers with LRP6E1E2. (**A**–**D**) Molecular docking models of AptLRP6-A1~AptLRP6-A4 with LRP6E1E2 and details of the binding sites. The right figure shows the specific binding sites of the candidate aptamers with LRP6E1E2, in which the amino acid residue is shown as stick models in yellow, and the nucleotide is shown as stick models in red.

**Figure 5 molecules-28-03838-f005:**
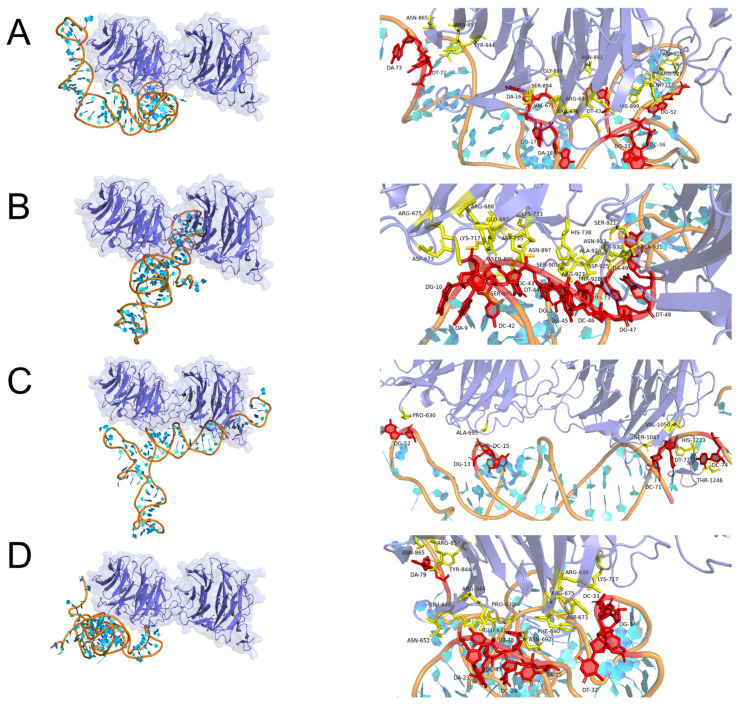
Docking prediction of candidate aptamers with LRP6E3E4. (**A**–**D**) Molecular docking models of AptLRP6-A1~AptLRP6-A4 with LRP6E3E4 and details of the binding sites. The right figure shows the specific binding sites of the candidate aptamers with LRP6E3E4, in which the amino acid residue is shown as stick models in yellow, and the nucleotide is shown as stick models in red.

**Figure 6 molecules-28-03838-f006:**
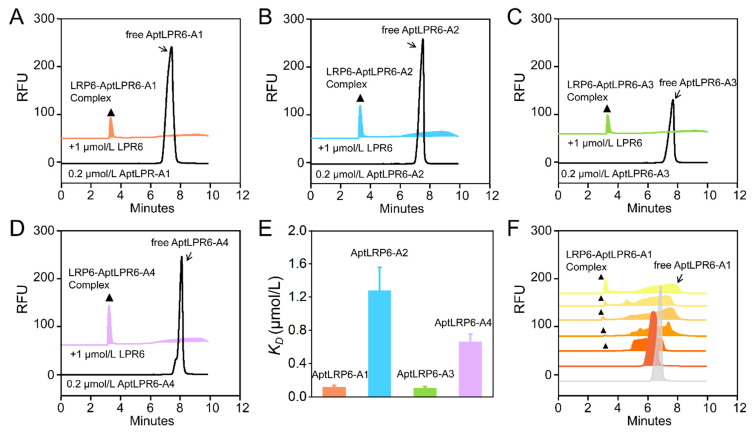
Candidate aptamers’ affinity evaluation and complex formation. (**A**–**D**) The binding of AptLRP6-A1~AptLRP6-A4 to the target protein LRP6. (**E**) Equilibrium dissociation constants of candidate aptamers. (**F**) CE analysis of complex formation process. The spectral lines from bottom to top are: 0.2 μmol/L AptLRP6-A1, 0.2 μmol/L AptLRP6-A1 + 0.01 μmol/L LRP6, +0.02 μmol/L LRP6, +0.05 μmol/L LRP6, +0.1 μmol/L LRP6, +0.2 μmol/L LRP6, and +0.5 μmol/L LRP6. In all of them, the arrow points to the free aptamers, and the “▲” points to the complex of candidate aptamers and LRP6.

**Figure 7 molecules-28-03838-f007:**
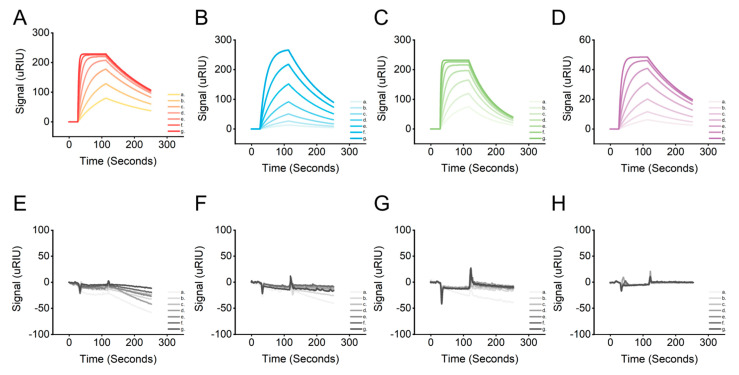
SPR characterizes candidate aptamers’ affinity and specificity. (**A**–**D**) The SPR response spectrums of the binding of AptLRP6-A1~AptLRP6-A4 to LRP6. (**E**–**H**) The SPR response spectrums of the binding AptLRP6-A3 to HSA, LDL, FZD8, and S100B. (a–g) The aptamer concentrations of 0.015625 μmol/L, 0.03125 μmol/L, 0.0625 μmol/L, 0.125 μmol/L, 0.25 μmol/L, 0.5 μmol/L, and 1 μmol/L, respectively.

**Figure 8 molecules-28-03838-f008:**
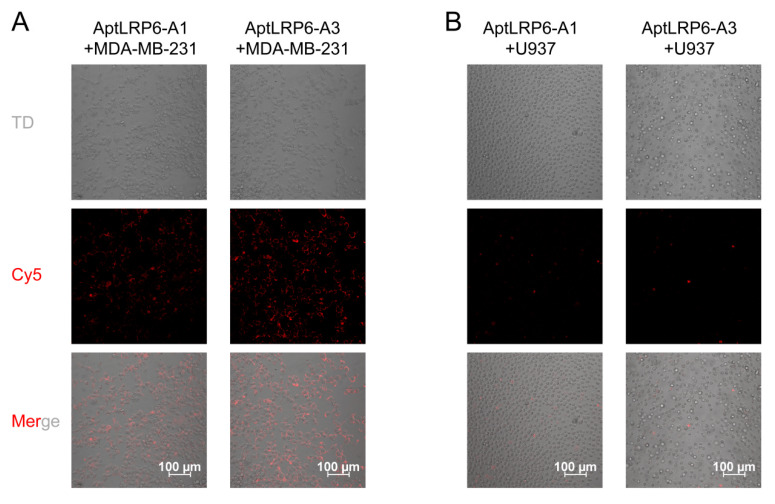
Binding of candidate aptamers to target cells. At the cellular level, AptLRP6-A3 showed the highest affinity for target cells. (**A**) Imaging of candidate aptamers AptLRP6-A1 and AptLRP6-A3 after co-incubation with MDA-MB-231 cells. (**B**) Imaging of candidate aptamers AptLRP6-A1 and AptLRP6-A3 after co-incubation with U937 cells.

**Table 1 molecules-28-03838-t001:** Four candidate aptamer sequences and thermodynamic parameters.

Name	Sequence (5′-3′)	Frequency	ΔG (kcal/mol)	ΔH (kcal/mol)	ΔS (cal/K·mol)	Tm (°C)	*K_D_* (μmol/L)
AptLRP6-A1	P1-CCGCAGGCAGCTGCCATTAGTCTCTATCCGTGACGGTATG-P2	1161	−7.29	−95.90	−285.7	62.5	0.118 ± 0.019
AptLRP6-A2	P1-GCCACATTAGTCTCACCACTACCTGCGTACCTACCGCCGC-P2	306	−1.78	−55.00	−171.5	47.3	1.279 ± 0.279
AptLRP6-A3	P1-GCAGCTAAGCAGGCGGCTCACAAAACCATTCGCATGCGGC-P2	133	−3.07	−83.00	−257.7	48.9	0.105 ± 0.018
AptLRP6-A4	P1-CGACTTGCCTATCGGCATGACACAATCTTTTGGAGCGTAA-P2	45	−1.54	−51.80	−162	46.5	0.661 ± 0.092

Note: P1 and P2 are the forward and reverse primers of the candidate aptamer sequences, respectively. P1: 5′-AGCAGCACAGAGGTCAGATG-3′; P2: 5′-CCTATGCGTGCTACCGTGAA-3′.

**Table 2 molecules-28-03838-t002:** The binding kinetic parameters of candidate aptamers and LPR6 were determined via SPR.

Name	Bmax ([Signal (uRIU)])	ka (1/(M · s))	kd (1/s)	*K_D_* (mol/L)	U-Value: ka/kd (%)
AptLRP6-A1	231.71	4.12 × 10^5^	5.53 × 10^−3^	1.34 × 10^−8^	10
AptLRP6-A2	315.33	4.54 × 10^4^	7.88 × 10^−3^	1.73 × 10^−7^	6.3
AptLRP6-A3	237.77	5.02 × 10^5^	1.29 × 10^−2^	2.57 × 10^−8^	>50
AptLRP6-A4	50.93	1.28 × 10^5^	6.51 × 10^−3^	5.10 × 10^−8^	9.4

## Data Availability

Not applicable.

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
