# Peer review of "Screening and Identification of ssDNA Aptamers for Low-Density Lipoprotein (LDL) Receptor-Related Protein 6"

_molecules, 2023, doi:10.3390/molecules28093838_

Round 1

Reviewer 1 Report

The paper “Screening and identification of ssDNA aptamers for low-density lipoprotein (LDL) receptor-related protein 6” by Xiaomin Zhang and coauthors describes the development of the ssDNA aptamers as high specific recognition element for lipoprotein receptor-related protein 6 (LRP6). They succeeded in obtaining two aptamers possessing relatively low KD values of 0.118 and 0.105 μM. The effectiveness of the aptamers as recognition element has been shown in experiments on specific binding to the cell surface. The work is well done, the results are beyond doubt. The obtained material is promising for further application in the diagnosis and treatment of the corresponding diseases.

Comment.

Please check the text carefully to avoid misunderstandings, e.g.

line 104, caption to Fig. 1,  written “….B represents blank, C represents the ssDNAN40 control…” should be “….B represents blank, D represents the ssDNAN40 control…” – there is no C column in gel electrophoresis diagram.

To summarize, the work is of interest for specialists and can be accepted for publication after a small revision.

Reviewer 2 Report

Huang et al reported a selection of aptamers against LRP6 protein using the CE-SELEX method. After identifying few candidates, the authors tried to understand aptamers’ binding epitope using NUPACK and molecular docking simulation. They then characterized the selected aptamers using SPR, CE and gold NPs assay. Lastly, they confirmed the binding of the selected aptamers at the cellular level. Given that there is no existing aptamer to date against LRP6 protein, I recommend a major revision for this manuscript.

Comments:

-       The authors should give more information about LPR6 protein in the introduction, especially the physiological range of the protein between normal and disease states. it would be helpful for readers to evaluate the potential in using the selected aptamers.

-       What is the selection condition? What the concentration of ions/salts in selection buffer? The authors should comment on those points because DNA aptamers are known to function/bind differently depending on the buffer environment.

-       The authors then should perform the binding assay under the same conditions as they used for selection and also physiological condition.Would it be a difference between 2 conditions?

-       The authors identified few key residues for binding using molecular  docking. Would it be possible to validate the hypothesis? Perhaps, mutate the aptamers selectively at those residues and perform binding assay.

-       Kd of selected aptamer is at high nM- low uM range. Would it be enough to detect LPR6 protein at physiological concentration?

-       Figure 6: please re-organize the panels in the conventional order.

-       Please show all graph (similar to the one shown in Figure 6-F for all aptamer candidates). Perhaps put them in the SI.

-       Figure 8. Please show zoom-in cell images. LRP6 is a trans-membrane protein so I guess it would be possible to see the binding of aptamers near the plasma membrane/cell surface. With the current images shown in Figure 8, it is very difficult for the readers to see and evaluate the results.

-       Line 191: please check grammar.

Round 2

Reviewer 2 Report

The authors have addressed my comments. The manuscript can be recommended for publication.